# No evidence for a trade-off between reproduction and survival in a meta-analysis across birds

**Lucy A Winder\*, Mirre JP Simons, Terry Burke**

Ecology & Evolutionary Biology, School of Biosciences, The University of Sheffield, Sheffield, United Kingdom

## eLife Assessment

This **important** study challenges conventional life-history theory by demonstrating that reproductive-survival trade-offs are minimal in birds, except when reproductive effort is experimentally exaggerated. The evidence is **solid**, drawing from a meta-analysis of over 30 bird species, and effectively separates the effects of individual quality from reproductive costs. The findings will be of broad interest to evolutionary biologists and ecologists studying life-history trade-offs and reproductive strategies.

**\*For correspondence:**
lucy.anne.winder@gmail.com

**Competing interest:** The authors declare that no competing interests exist.

**Abstract** Life-history theory, central to our understanding of diversity in morphology, behaviour, and senescence, describes how traits evolve through the optimisation of trade-offs in investment. Despite considerable study, there is only minimal support for trade-offs within species between the two traits most closely linked to fitness – reproductive effort and survival – questioning the theory's general validity. We used a meta-analysis to separate the effects of individual quality (positive survival/reproduction correlation) from the costs of reproduction (negative survival/reproduction correlation) using studies of reproductive effort and parental survival in birds. Experimental enlargement of brood size caused reduced parental survival. However, the effect size of brood size manipulation was small and opposite to the effect of phenotypic quality, as we found that individuals that naturally produced larger clutches also survived better. The opposite effects on parental survival in experimental and observational studies of reproductive effort provide the first meta-analytic evidence for theory suggesting that quality differences mask trade-offs. Fitness projections using the overall effect size revealed that reproduction presented negligible costs, except when reproductive effort was forced beyond the maximum level observed within species, to that seen between species. We conclude that there is little support for the most fundamental life-history trade-off, between reproductive effort and survival, operating within a population. We suggest that within species the fitness landscape of the reproduction–survival trade-off is flat until it reaches the boundaries of the between-species fast–slow life-history continuum. Our results provide a quantitative explanation as to why the costs of reproduction are not apparent and why variation in reproductive effort persists within species.

## Introduction

Across taxa, we see wide variation in life-history traits, such as the number of offspring produced and time spent raising young (*Lack, 1947*; *Harvey and Clutton-Brock, 1985*; *Ricklefs, 2000*). The central idea in life-history theory is that resources are finite, forcing trade-offs, meaning that investment in one aspect of life requires the sacrifice of another (*Stearns, 1976*; *van Noordwijk and de Jong,*

*1986*; *Kirkwood and Rose, 1991*; *Lemaître and Gaillard, 2017*). As reproduction is one of the most resource-demanding life stages, it is expected that within-species variation in offspring production will be constrained by the cost of producing and raising young. It is thought that the fitness costs of reproduction are largely incurred as a decrement to parental survival, explaining why a species evolved a certain amount of reproductive effort, whilst also explaining the fast–slow life-history continuum between reproduction and lifespan observed across species (*Kirkwood and Rose, 1991*). As reproduction and survival are the two components of life-history most closely related to fitness, this central trade-off has been the subject of much theoretical and empirical research, branching fields ranging from ecology, evolutionary biology to ageing research (*Reznick, 1992*; *Kirkwood and Austad, 2000*; *Flatt and Partridge, 2018*).

Brood size manipulations of birds in natural conditions have provided arguably the best experimental paradigm in which to test the survival costs of reproduction as they directly alter parental care investment. Experimental increases in brood size result in increased parental effort, suggesting that parents can typically cope with increased reproductive demands (*Pettifor et al., 1988*; *Lessells, 1986*; *Monaghan and Nager, 1997*; *Conrad and Robertson, 1992*; *Hegner and Wingfield, 1987*). However, the expected increased costs of reproductive effort are not always detected and the current estimate of the cost to reproduce across studies suggests only a small and inconsistent effect (*Reznick, 1985*; *Zera and Harshman, 2001*; *Santos and Nakagawa, 2012*; *Cohen et al., 2020*). The absence of a cost of reproduction (determined through experimental studies) on survival means that costs must arise elsewhere or, alternatively, that individuals may differ in quality (determined through observational studies). By quality, we refer to individuals operating at their own maximum reproductive output, determined by their phenotypic condition, local or temporal genetic adaptation, and the surrounding environment, and should manifest as a positive correlation between two fitness-associated traits (such as reproduction and survival) (*Pettifor et al., 1988*; *Cohen et al., 2020*; *Charnov and Krebs, 1974*; *Wilson and Nussey, 2010*; *Drent and Daan, 2002*). The relative importance of the trade-off between reproductive effort and survival to life-history theory and the biology of ageing (*Kirkwood and Austad, 2000*), therefore, remains unclear. In addition, the compelling theoretical explanation for the lack of an apparent trade-off due to the confounding effects of individual quality has not often been investigated on a quantitative level (*van Noordwijk and de Jong, 1986*; *Cohen et al., 2020*; *Descamps et al., 2016*; *Vedder and Bouwhuis, 2018*). Our study aims to separate the effects of a trade-off between reproductive effort and parental survival and the effects of individual quality by comparing studies of brood manipulations and naturally varying clutch size.

Here, we present a meta-analysis that distinguishes between individual quality effects and the costs of reproduction. To do this, we tested how parental annual survival in birds is affected by the clutch size they cared for using brood manipulation studies (costs of reproduction) and observational studies of natural variation in clutch size (individual quality effects) (note brood size and clutch size are used interchangeably throughout this article). We find the effect of quality is associated with higher survival chances given an increase in reproductive effort, and that this effect is opposite but equal in magnitude to the costs of reproduction. Our analysis also allowed us to transform the response variable, scaling for variance and mean, given that a per-egg increase in clutch size does not equate to the same proportional increase in reproductive effort for all species equally. Our findings suggest that species that generally lay smaller clutches are affected more severely by brood size manipulations.

To predict the evolutionary consequences of the effect sizes that we estimated using the meta-analysis, we projected the fitness consequences for a change in clutch size life-history strategy. We found that the costs to parental survival as estimated from brood size manipulation studies translate into negligible fitness costs, with a relatively flat fitness landscape, suggesting that birds underproduce in terms of clutch size, given the absence of fitness costs. Our results therefore suggest that, in wild populations, parental survival costs are, at most, a small component of the total fitness costs of investing parental effort. We therefore infer that, though the survival–parental care trade-off does exist within species, it not strong enough to constrain clutch size and can therefore not explain clutch size evolution and thus variation within species.

## Results

The relationship between clutch size and survival was significantly different and opposite between observational and brood manipulation studies, irrespective of how clutch size was scaled ($p < 0.01$,

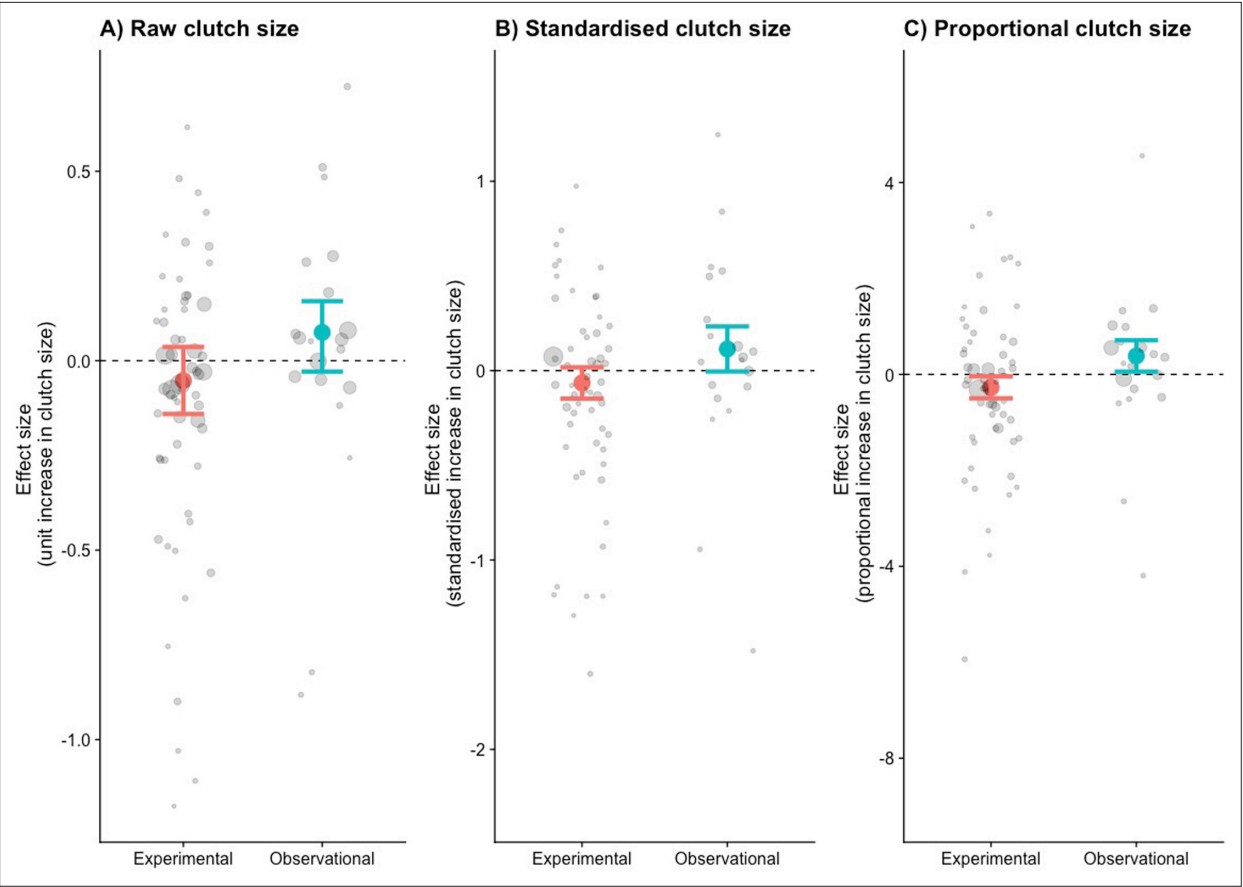

**Figure 1.** The effects size (log odds of survival given an increase in clutch size) for three different measures of clutch size. (**A**) Raw, (**B**) standardised, and (**C**) proportional clutch size. Coloured points are the combined effect sizes of the odds ratios with their 95% confidence intervals (n = 58 experimental and 20 observational). Points are coloured by whether they represent brood manipulation experiments (costs of reproduction) or they are observational (quality). Grey points are the odds ratios of each study, with their sizes weighted by the sampling variance.

The online version of this article includes the following figure supplement(s) for figure 1:

**Figure supplement 1.** Survival effects for increasing clutch size for female, male, and mixed-sex studies.

**Figure supplement 2.** Funnel plot of meta-analysis residuals against standard error.

*Figure 1*, *Table 1*). Within natural variation (observational studies), parents with larger clutches showed increased survival (or parents with smaller clutches showed reduced survival). In contrast, when broods were experimentally manipulated, the opposite relationship was found; increasing brood sizes decreased survival. Although the difference in overall effect size between experimental and natural variation in clutch size was strongly significant in each comparison made, the individual overall effect sizes only became significantly different from zero when clutch size was expressed as a proportional increase. Expressing clutch size as a proportional increase corrects for the variation in average clutch size observed across the species included in this analysis, which ranged from 2 to 11. The parental effort required to raise two instead of one chick is potentially doubled, whereas one additional chick in a brood of 11 is likely to require only a marginal increase in effort. Indeed, also when using a between-species comparison, the effects of clutch size manipulation and quality were strongest in the species that laid the smallest clutches, suggesting that costs to survival were only observed when a species was pushed beyond its natural limits (*Figure 2*, *Figure 2—figure supplement 1*, *Appendix 1—table 1*).

Males and females did not differ in their survival response to changing clutch size (*Appendix 1 - table 2*, *Figure 1—figure supplement 1*, contrary to *Santos and Nakagawa, 2012*). The variance assigned to the random effects in the model was largely accounted for by study (*Appendix 1—table 3*). Species identity accounted for more variation than the phylogeny, indicating that species vary in

**Table 1.** Effect size estimates for the odds of survival with increasing clutch size (raw, standardised, and proportional clutch size). The p-values indicate the difference between brood manipulations and observational data, with the individual effect p-values (from zero) in parentheses. Values in bold show statistical significance of p < 0.05.

| | Parameter | | Effect size | 95% CI lower bounds | 95% CI upper bounds | p Value | p Value (Individual) |
|---|---|---|---|---|---|---|---|
| Raw | Clutch size | Brood manipulation | –0.0522 | –0.1406 | 0.0363 | **0.0007** | (0.2477) |
| | | Observational | 0.0747 | 0.1571 | –0.0288 | | (0.1571) |
| Standardised | Clutch size | Brood manipulation | –0.0651 | –0.1478 | 0.0177 | **0.0065** | (0.1232) |
| | | Observational | 0.1143 | –0.0046 | 0.2333 | | (0.0595) |
| Proportional | Clutch size | Brood manipulation | –0.2703 | –0.4984 | –0.0423 | **0.0005** | **(0.0202)** |
| | | Observational | 0.3850 | 0.0583 | 0.7116 | | **(0.0209)** |

Model = ~obervational_or_experimental, random = (species, phylogeny, study reference).

the relationship between survival and reproductive effort, irrespective of their shared evolutionary ancestry. However, our dataset included few closely related species, which reduces our ability to estimate phylogenetic effects (*Figure 3*).

## Projected fitness consequences of the costs of reproduction

From our meta-analysis, we now have a quantifiable and comparable effect size for the survival costs of reproduction that we can use to predict the evolutionary consequences of individuals in a given population increasing their clutch size across a range of life histories. To this end, we projected the fitness consequences of increased reproductive effort, starting with the average effect size estimate per egg (–0.05, *Table 1*) across a range of life histories, for a range of annual survival rates and clutch sizes (*Figure 4*). Overall, the effect size estimated in the meta-analysis (–0.05) resulted in a gain of fitness when reproductive output increased, especially in hypothetical species with low survival and small clutches. If we used a stronger effect size (–0.15, the lower confidence interval of our meta-analysis),

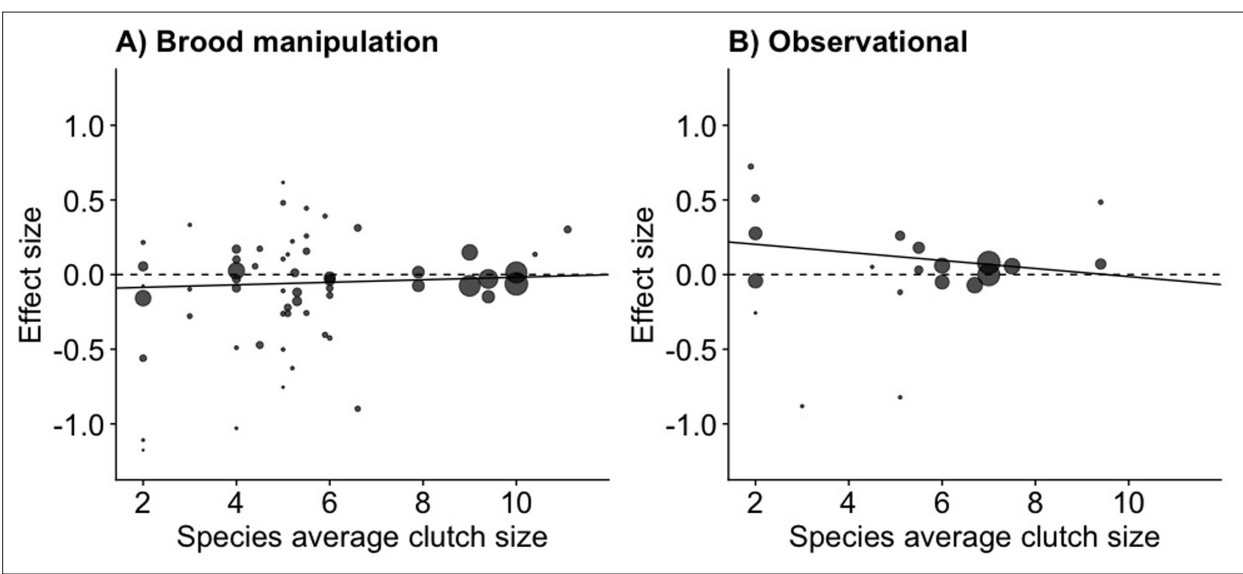

**Figure 2.** The meta-analytic linear regression (*Appendix 1—table 1*) of the effect size for increasing clutch size (per egg) on parental survival, given the average clutch size for the species for (**A**) brood manipulation and (**B**) observational studies. Species with small clutch sizes showed stronger costs of reproduction and a stronger relationship with quality (Interaction between treatment and species clutch size effect size = –0.036, p=0.015). The points are the survival effect sizes (log odds ratio) per egg (as in *Figure 1A*) on parental survival in each study, with the point size reflecting the meta-analytic weight of that study (n = 58 experimental and 20 observational).

The online version of this article includes the following figure supplement(s) for figure 2:

**Figure supplement 1.** Brood size comparisons across studies.

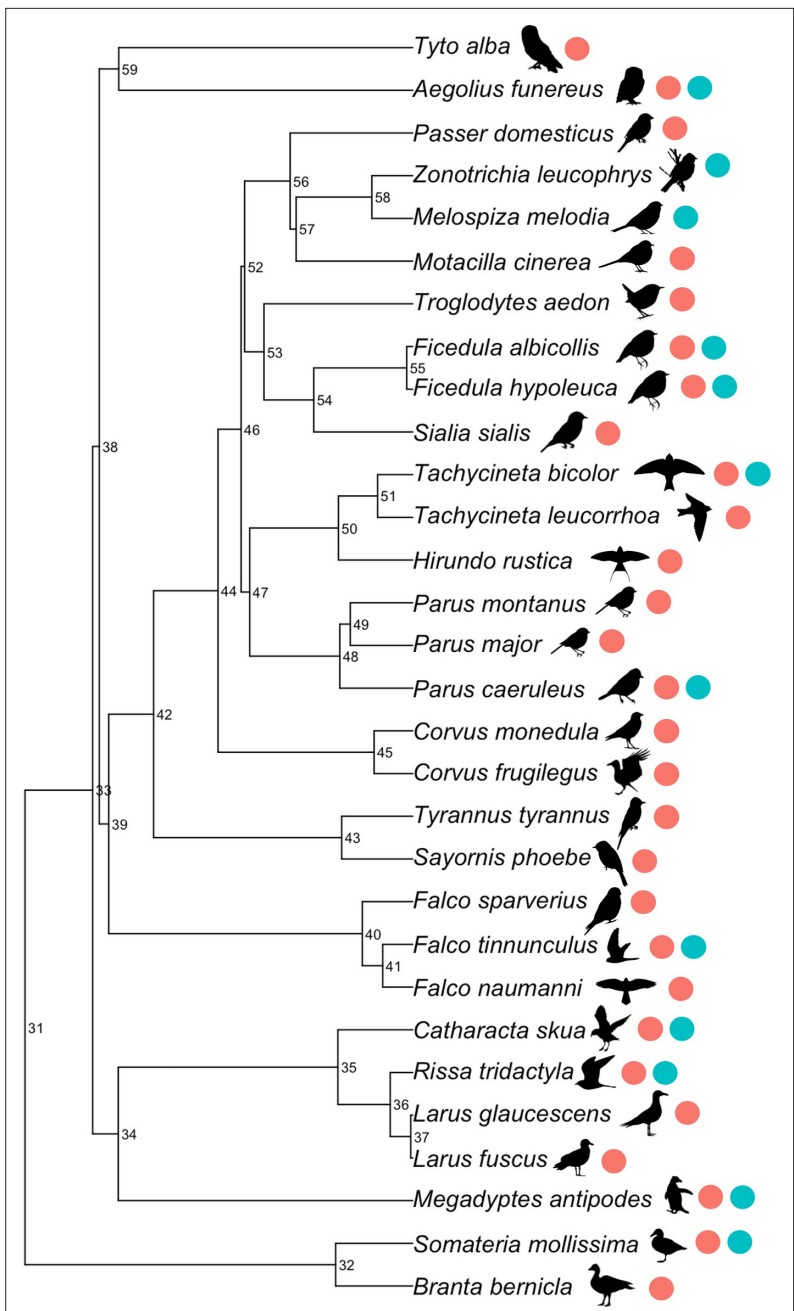

**Figure 3.** Phylogenetic tree of species included in our meta-analysis. Numeric labels denote the node support (as a percentage). Coloured points indicate whether the species were included in the experimental or observational studies.

this led to a reduction in fitness in almost all cases. Conversely, the benefit of higher reproductive output was largely offset by the cost of survival when a species' survival rate and clutch size were high. When we increased the effect size up to fivefold, fitness costs of reproduction became more pronounced, but were still not present in species with small clutches and short lifespans.

Under long-term evolution, these selection differentials should lead to individual hypothetical species moving towards the diagonal (bottom-left to the right-top corner). This diagonal represents the observed fast–slow pace of life continuum observed among species (*Healy et al., 2019*). Exemplar species (i.e., with survival and average clutch size combinations observable in wild populations), for which we predicted the fitness consequences of the costs of reproduction, lie on this comparative

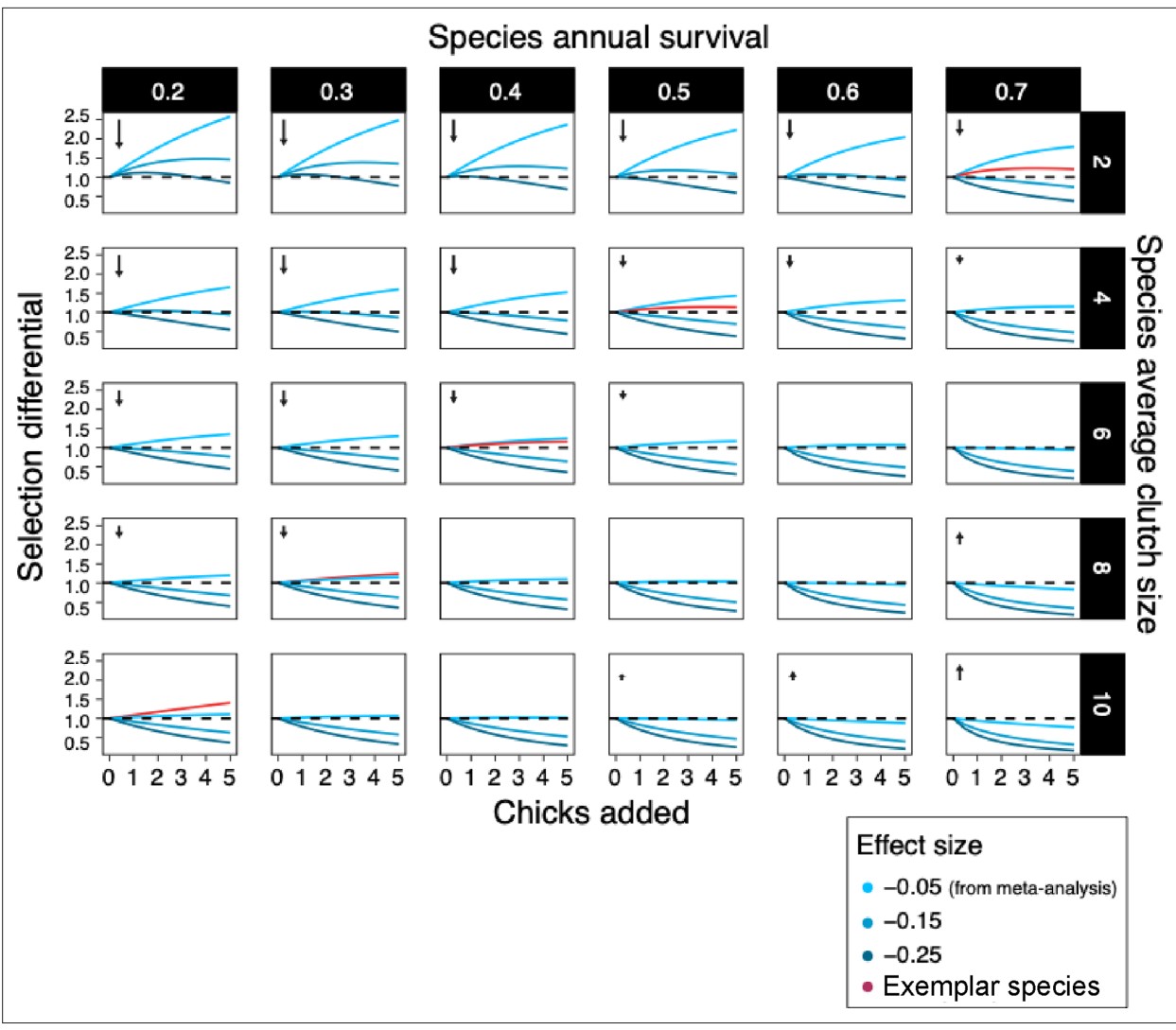

**Figure 4.** Isoclines of selection differentials among hypothetical control populations (in which individuals reproduce at the species' mean rate) and hypothetical brood-manipulated populations (where individuals reproduce at an increased rate compared to control) for their whole lives. Selection differentials (i.e., the relative difference in lifetime reproductive output between hypothetical control and brood-manipulated populations) above 1 represents high lifetime fitness. Survival rates, clutch sizes, the magnitude of the manipulation (chicks added) and effect sizes represent the range of these variables present in the studies used in our meta-analysis. For each clutch size, we used a predicted survival rate and effect size to give isoclines that are biologically meaningful (exemplar birds shown in red). Arrows indicate the relative size and direction of selection in life-history space (on the reproduction axis). The costs of reproduction we estimated within species are predicted to result in a fast–slow life-history continuum across species, and the exemplar species we used as examples fit on this diagonal of survival rate/ clutch size combinations. We suggest that individual species show limited costs of reproduction as they operate within relatively wide constraints imposed by the cost of reproduction that is responsible for the strong life-history trade-off observed across species.

diagonal in life-history. In these exemplar species, the selection differential was observed to lie slightly above one, indicating that individuals having a higher clutch size than the species' average would gain a slight fitness benefit. The fitness costs and benefits did, in general, not diverge substantially with the addition of chicks, but flattened, suggesting that the costs of survival counterbalance the benefits of reproduction across a range of reproductive outputs within a species.

The low costs of reproduction that we estimated could still be responsible for between-species life-history evolution, constraining species reproductive output and survival combination to fall along the diagonal of the fast–slow pace of life continuum. How selection pressures translate into short-term and longer-term evolutionary trajectories is uncertain. Often directional selection estimated in the wild does not translate to the inter-generational change on the population level (*Pujol et al., 2018*). Note, however, that only far away from the diagonal did our fitness projections reach a magnitude

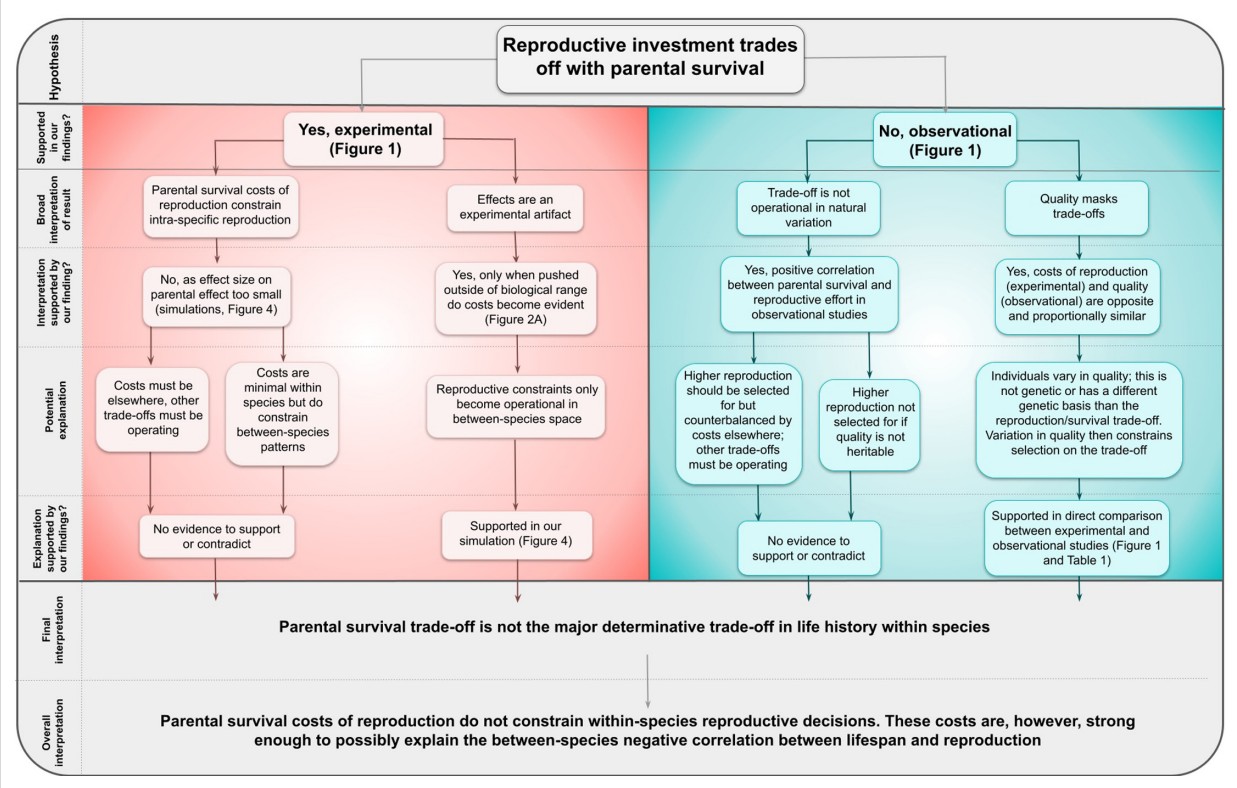

**Figure 5.** Decision tree representing the logical steps from our original hypothesis to our overall interpretation of our findings.

that would be predicted to lead to rapid evolutionary change (*Gingerich, 2009*; see Appendix 2). The weak selection effects that lie on the diagonal are probably to be counterbalanced in the wild by factors such as environmental effects and genetic effects (e.g., gene flow from immigration or random mutation) (*Postma et al., 2007*). We argue that within-species the minimal costs of reproduction, a flat fitness landscape and quality effects (*Figure 1*) together explain why individuals appear to under-produce. Only when individuals are pushed beyond the observed between-species constraint do costs become apparent (*Figures 1 and 2*).

Our interpretation of the reproduction/lifespan life-history trade-off, based on our quantitative meta-analysis and subsequent fitness projections, explains several key observations and contradictions in the field. A strong trade-off is observed between species, but within species this trade-off is not apparent and variation in reproductive output is maintained within fitness boundaries similar to those that determine the between-species life-history trade-off. The implication of this conclusion is that the costs of reproduction are likely to operate on a physiological level, but that the fitness consequences will remain largely flat over a species' observed variation in reproductive output. These effects are further obscured by the effects of quality (observational studies), which are opposite in sign and magnitude to the cost of reproduction (experimental studies) (*Figure 1*) and are likely to further flatten the fitness landscape.

## Discussion

Our results provide the first meta-analytic evidence that differences in individual quality drive variation in clutch size and that parental survival costs do not constrain within-species variation in reproductive effort (*Figure 5*). Here, we use the definition of quality as a combination of traits that give an individual higher fitness (*van Noordwijk and de Jong, 1986*). The finding of individual quality being a driver of variation in clutch size is not necessarily a surprising one, even though this contradicts the general theory that trade-offs drive variation in clutch size, as among-individual heterogeneity is also a well-established effect across populations in the field of ecology and evolution (such as age-specific reproductive performance) (*van Noordwijk and de Jong, 1986*; *Vedder and Bouwhuis, 2018*). There

has previously been discussion of why positive genetic correlations have been found in the wild and result from, for example, positive pleiotropy (*Maklakov et al., 2015*). Furthermore, empirical data has also found an absence of a trade-off between reproduction and survival (*Reznick et al., 2004*; *Žák and Reichard, 2021*; *Kimber and Chippindale, 2013*). A recent meta-analysis on genetic life-history trade-offs found several positive genetic correlations between survival and other life-history traits, which is in conflict with life-history trade-off theory (*Chang et al., 2024*). Brood manipulations often fail to take into account that variation in resource acquisition may be larger than variation in resource allocation, and though intuitively manipulation of effort tests variation in resource allocation, it fails to account for the baseline variation in reproductive performance (*van Noordwijk and de Jong, 1986*). This presents a fundamental flaw which our results demonstrate – variation in resource allocation can be present and testable through experiments (i.e. brood manipulations) but may only account for a small proportion of variation observed in reproductive output. Perhaps fitting with this interpretation or with additional costs from recurring bouts of enhanced reproductive effort, Jackdaws (*Corvus monedula*) show costs of increased brood size only when applied sequentially across years, but note that for this study there is no control group that is only manipulated in single years (*Boonekamp et al., 2014*).

The reason selection is not acting on 'high-quality' individuals is currently unknown, but it is likely that environmental variability leads to alternative phenotypes being selected for at different points in space and time (also discussed in *Pujol et al., 2018*). It is also possible that the quality effect could represent a terminal effect, where individuals have lower reproductive output in the year preceding their death, thereby driving the trend for naturally lower laying birds to have lower survival when estimated between individuals (e.g. *Coulson and Fairweather, 2001*; *Rattiste, 2004*; *Simons et al., 2016*; *Coulson and Fairweather, 2001*; *Rattiste, 2004*; *Simons et al., 2016*; also see *Hammers et al., 2012* for age-related changes in reproductive output). The effect on parental annual survival of having naturally larger clutches was significantly opposite to the result of increasing clutch size through brood manipulation and quantitatively similar. Parents with naturally larger clutches are thus expected to live longer, and this counterbalances the 'cost of reproduction' when their brood size is experimentally manipulated. It is, therefore, possible that quality effects mask trade-offs. Furthermore, it could be possible that individuals that lay larger clutches have smaller costs of reproduction, that is, would respond less in terms of annual survival to a brood size manipulation, but with our current dataset we cannot address this hypothesis (*Figure 5*). The effect of a change in clutch size on parental survival may also be non-linear, but it remains to be determined what shape, if not linear, the relationship would be. Although these possible non-linear effects warrant investigation, these relationships likely differ between species and inclusion in our current work here could lead to spurious relationships being reported.

For both costs of reproduction and quality effects, we found that species that laid the smallest clutches showed the largest effects. Brood manipulations that affect parental survival are thus likely to be the result of pushing parental effort beyond its natural optima (*Figure 2—figure supplement 1*) and so arguably that brood manipulations are not necessarily a good test of whether trade-offs happen in the wild. The classic trade-off between adult survival and the clutch size cared for is only apparent when an individual is forced to raise a clutch outside of its individual optimum, and these effects are confounded or even fully counterbalanced by differences in quality (as theorised in *van Noordwijk and de Jong, 1986*).

Our fitness projections of the consequences of the costs of reproduction using the overall effect size we estimated suggest that, for current extant species, the within-species fitness landscape of the reproduction survival trade-off is flat. Species' life-history decisions are constrained within a broader fast–slow life-history continuum, explaining why variation within species in reproductive effort, such as in clutch size, is large and near universal. Of course, this analysis does not fully explore more complex variation in clutch size observed in the wild, such as that observed between temperate and tropical reproductive strategies. Such work would prove useful in understanding why variation in clutch size is maintained across species with different life-history strategies. Our results do, however, provide a key step in understanding the generalised relationship between reproductive effort and survival across species (*Figure 5*). Our interpretation also assumes that other fitness costs of reproduction are smaller or at least less relevant than survival costs. However, it is possible that such costs are actually more important, and it should be noted that effects such as those on offspring quality (e.g. *Conrad*

*and Robertson, 1992* and *Smith et al., 1989*), parental condition other than survival (e.g. *Reid, 1987* and *Kalmbach et al., 2004*), and future reproductive effort e.g. *Järvistö et al., 2016* have been observed (also see *Figure 5*). However, the importance of these effects is likely to vary considerably in different species. Using offspring quality as an example, some species produce sacrificial offspring, others experience catch-up growth, meaning that though the effect of increasing offspring number on offspring quality is important for certain species, drawing generalised across-species conclusions is unlikely to be possible. Conversely, the survival cost to reproduce is thought to be universal across both bird species and across taxa, and it is perhaps for this reason that the reproduction–survival trade-off has been considered to contribute more to variation in reproductive effort than other trade-offs. Our work refutes this idea directly. Indeed, the few studies that have measured the different domains that contribute to fitness in brood size manipulation studies concluded that only in combination do these costs result in balancing selection for the current most common brood size in the population (*Daan et al., 1990*; *Verhulst and Tinbergen, 1991*; *Tinbergen and Daan, 1990*). Such classic trade-off explanations do, however, fail to explain why variation in reproductive effort is prevalent within species and why between-species life-history trade-offs appear so much stronger and conclusive. Our analysis and interpretation suggest that, at its optimum, the within-species trade-off between survival and reproduction is relatively flat, and thus neutral to selection (supporting the theory presented in *Cohen et al., 2020*). We suggest that the lack of evidence supporting trade-offs driving within-species variation does not necessarily mean that physiological costs of reproduction are non-existent (e.g. *Smith et al., 1989* and *Lemaître et al., 2015*), but rather that, within the wild and within the natural range of reproductive activities, such costs are not relevant to fitness. One key explanation for this effect supported by our meta-analysis (*Figure 5*) and prior theory (*van Noordwijk and de Jong, 1986*) is that individuals differ in quality.

## Methods
### Study sourcing and inclusion criteria

We extracted studies of parental survival to the following year given clutch size raised using the following inclusion criteria similar to *Santos and Nakagawa, 2012*: the study must be on a wild population; the study must contain variation in the number of offspring produced/raised (hereafter referred to as clutch size for simplicity), the study must report variation in clutch size in relation to parental survival to the following year (including both experimental and observational studies) and must provide sample sizes. We did include studies where parental survival was reported for both parents combined as opposed to *Santos and Nakagawa, 2012*, who required male and female data to be reported separately. Excluded studies and the grounds for their removal are given in the supplementary information (*Appendix 1—table 4*). We started by, first, extracting data (clutch size raised after manipulation and associated parental survival to the next year) from the included brood manipulation studies from *Santos and Nakagawa, 2012* and then searched the literature to include more recently published studies (Appendix 2), and in this search also added studies Santos and Nakagawa missed. In addition to brood manipulation studies, we extracted data from studies that correlated variation in parental survival with natural variation in clutch size (observational studies). For each species used in the brood manipulation studies, we aimed to find the same species for the observational studies to ensure that the effects of quality were estimated across a similar range of species. The reason for this is to ensure the experimental and observational datasets are comparable in terms of species, which could bias results if not considered, and so facilitate a more direct comparison between the quality effect (observational studies) and the trade-off (experimental studies). Where there was no equivalent study in the same species, we attempted to find a study of a congener. In most cases, observational data were obtained from either the same paper as the one describing brood manipulations (11 studies) or via searching for other papers by the same authors (2 studies). If this failed to produce observational data, a search was conducted following the same protocol as for the brood manipulation experiments, but also specifying species, genus, and/or common name in the search (seven studies). Of the 28 species used in the brood manipulation studies, we were able to find 10 of the same species in observational studies. We were able to find a further six species which were congeners for observational studies.

From the literature search, 78 individual effect sizes from 30 species and 46 papers were used (20 observational and 58 experimental studies). While extracting these studies, we also made note of the average clutch size of the species and the within-species standard deviation in clutch size. We extracted this information from the paper containing the study, but if the information was missing, we searched other published literature with the aim to find the information from a similar population (i.e. at a similar latitude).

## Extracting effect sizes

We used the absolute value of offspring in the nest (for experimental studies, this is the number of offspring after the manipulation occurred) and associated parental survival to estimate an effect size by performing a logistic regression to obtain the log odds ratio for parental survival, given the clutch size (i.e. positive effect sizes indicate an increased chance of survival). For example, a bird who laid a clutch size of 5 but was manipulated to have −1 chick was recorded to have a clutch size of 4. If the manipulation was reported but absolute value of offspring produced was not, we used clutch size to be the species average ± the number of offspring added or removed. Parental survival was modelled as a binary response variable for the number of birds who survived and who died after raising a given clutch size. Clutch size was averaged (mean) if a single estimate of survival was reported for multiple clutch sizes. 'Year' was included as an explanatory variable to correct for between-year variation in adult survival, where data were presented for multiple years. We standardised the clutch size (by the mean of the species and by the within-species standard deviation in clutch size) and transformed clutch size to a proportion of the species mean. For species that have no within-species variance in clutch size, we used a value of 0.01 for the standard deviation in clutch size to prevent issues in calculations when using zero. We, therefore, expressed variation in clutch size in three ways: a raw increase in clutch size, a standardised clutch size, and a proportional clutch size.

## Meta-analysis

We ran a single model using the log odds ratio calculated for each clutch size transformation (i.e. raw, standardised, and proportional) to determine the effect of parental survival, given an increase in reproductive effort using the *metafor* package (*Viechtbauer, 2010*) in R 3.3.2 (*R Development Core Team, 2009*). From these models, we were also able to directly compare the effect size of brood manipulation studies and observational studies by including a categorical fixed effect for study type (i.e. experimental or observational). We included phylogeny as a correlation matrix in these meta-analytic models to correct for shared ancestry. The phylogeny was obtained using BEAST to measure a distribution of 1000 possible phylogenetic trees of the focal 30 species extracted from BirdTree (*Rubolini et al., 2015*). We also included species and each studies' journal reference as random effects in the model. From these models, we calculated the proportion of variance explained by the phylogenetic effect (*Nakagawa and Santos, 2012*).

We then tested the effect of the species' mean clutch size on the relationship between parental survival and clutch size. We ran a single model with the mean-centred (from all species used in the meta-analysis) species' average clutch size in interaction with treatment (brood manipulation or observational). Species, phylogeny, and reference were also included as random effects to correct for the similarity of effect sizes within species and studies.

The difference in survival for the different sexes was modelled for each clutch size measure. Brood manipulation studies and observational studies were analysed in separate models. Sex was modelled as a categorical moderator (41 female studies, 27 male studies, and 10 mixed studies). Species, phylogeny, and reference were included as random effects (*Appendix 1—table 2* and *Figure 1— figure supplement 1*).

## Publication bias

Much of the data used in this analysis were taken from studies where these data were not the main focus of the study. This reduces the risk that our results are heavily influenced by a publication bias for positive results. A funnel plot for the survival against raw clutch size model is presented in supplementary information (*Figure 1—figure supplement 2*).

## Fitness projections

We calculated various isoclines using the brood manipulation overall effect size (–0.05, based on raw clutch size) that we estimated. Here, an isocline is a trendline representing the change in fitness returns over an individual's lifespan, given an increase in individual clutch size. An estimated 'lifetime reproductive output' was calculated for hypothetical control populations (starting with 100 individuals), where all individuals consistently reproduce at the level of a species mean and have a consistent annual survival rate. We calculated this 'lifetime reproductive output' (see Appendix 2 for equation details for calculating lifetime reproductive output) using combinations of a range of species' average clutch sizes at 2, 4, 6, 8, and 10 and survival rates of 0.2, 0.3, 0.4, 0.5, 0.6, and 0.7, which reflected the range of clutch sizes and survival rates seen in the species in our meta-analysis. We started with a hypothetical population size of 100 and calculated the populations' lifetime reproductive output rate that depends on how many offspring are produced in a clutch and how long individuals in the population live for. We could then use to model the effects of the costs of reproduction using the meta-analytic estimate of how brood size is associated with parental survival (through experimental studies).

The 'lifetime reproductive output' estimate was then repeated for a hypothetical population that reproduces at an increased level compared to control, that is, brood size enlargement, throughout their lives. These individuals therefore produce more offspring but face reductions to survival. This analysis allows us to determine if an overall fitness gain is achieved by producing more offspring despite paying an increased survival cost or whether the survival cost balances out the fitness gains of producing more offspring through reduced reproductive attempts. To obtain this, we added a range of 1–5 offspring to the clutch sizes of the control populations. Using a range of increased clutch sizes allowed us to investigate how increased reproductive effort would affect lifetime fitness. The survival costs were determined by the overall effect size found for brood manipulation studies (per egg). We modelled effect sizes of –0.05, –0.15, and –0.25, which represent, respectively, the meta-analytic overall effect size, its upper confidence interval, and a further severe effect within the observed effect sizes (rounded to the closest 0.05 for simplicity). For example, an individual who has an additional offspring in its nest would see a 5, 14, and 22% (respectively for each effect size) reduction in its survival odds compared to if it reproduced at its normal rate (i.e. the control population rate).

We then calculated the selection differential ($LRS_{brood\ manipulation}/LRS_{control}$) between the hypothetical control and 'brood manipulation' populations for each combination of survival rate, clutch size and effect size, and plotted this as an isocline. We further plotted the fitness consequences for five exemplar species, where survival rate and clutch size combinations are observable in the wild. We used effect sizes from model predictions at these survival rates and clutch size combinations rather than the meta-analytic mean, thereby providing a biological context.

## Acknowledgements

This work was supported by a Natural Environment Research Council (NERC) Adapting to the Challenges of a Changing Environment (ACCE) studentship to LAW, a Sir Henry Dale Fellowship to MJPS (Wellcome and Royal Society; 216405/Z/19/Z), and a grant from NERC (NE/J024597/1) to TB. For the purpose of open access, the author has applied a Creative Commons Attribution (CC BY) licence to any Author Accepted Manuscript version arising.

## Additional information

### Funding

| Funder | Grant reference number | Author |
|---|---|---|
| Wellcome Trust | 10.35802/216405 | Mirre JP Simons |
| Natural Environment Research Council | NE/J024597/1 | Terry Burke |
| Natural Environment Research Council | ACCE studentship | Lucy A Winder |

| Funder | Grant reference number | Author |
|---|---|---|

The funders had no role in study design, data collection and interpretation, or the decision to submit the work for publication. For the purpose of Open Access, the authors have applied a CC BY public copyright license to any Author Accepted Manuscript version arising from this submission.

## Author contributions

Lucy A Winder, Conceptualization, Data curation, Formal analysis, Investigation, Visualization, Writing – original draft, Project administration; Mirre JP Simons, Conceptualization, Supervision, Writing – review and editing; Terry Burke, Conceptualization, Supervision, Funding acquisition, Writing – review and editing

## Author ORCIDs

Lucy A Winder  https://orcid.org/0000-0002-8100-0568
Terry Burke  https://orcid.org/0000-0003-3848-1244

Reviewer #4 (Public review): https://doi.org/10.7554/eLife.87018.5.sa1
Author response https://doi.org/10.7554/eLife.87018.5.sa2

## Additional files

### Supplementary files
MDAR checklist

### Data availability
All data and code used in this study can be found at: https://doi.org/10.5061/dryad.q83bk3jnk.

The following dataset was generated:

| Author(s) | Year | Dataset title | Dataset URL | Database and Identifier |
|---|---|---|---|---|
| Winder L | 2023 | Data and code for: Optimal clutch size revisited: separating individual quality from the costs of reproduction | https://doi.org/10.5061/dryad.q83bk3jnk | Dryad Digital Repository, 10.5061/dryad.q83bk3jnk |

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

# Appendix 1

**Appendix 1—table 1.** Model outputs for meta-analyses estimating the effect size of the odds of survival for increasing clutch size given the species' average clutch size.

Treatment was coded as a categorical variable indicating whether studies were either experimental or observational. The species' average clutch size was centred to the average clutch size of all species used in the meta-analysis. The increase in clutch size was modelled as the raw increase in clutch size and the standardised increase in clutch size. We have not presented the proportional increase in clutch size as this represents the change from the species average and so is null in this model. Values in bold show statistical significance of $p < 0.05$.

| Model | Parameter | Effect size | 95% CI lower bounds | 95% CI upper bounds | p Value |
|---|---|---|---|---|---|
| | Intercept | −0.047 | −0.147 | 0.054 | 0.363 |
| | Treatment: observational | 0.150 | 0.079 | 0.222 | **<0.0001** |
| | Centred species clutch size | 0.011 | −0.012 | 0.034 | 0.337 |
| Raw | Treatment: observational × species clutch size | −0.036 | −0.066 | −0.007 | **0.015** |
| | Intercept | −0.065 | −0.222 | 0.092 | 0.418 |
| | Treatment: observational | 0.202 | 0.074 | 0.330 | **0.002** |
| | Species clutch size | 0.015 | −0.026 | 0.055 | 0.482 |
| Standardised | Treatment: observational × species clutch size | −0.057 | −0.117 | 0.002 | 0.057 |

Model = ~observational_or_experimental * mean_adjusted_clutchsize, random = (species, phylogeny, study reference).

**Appendix 1—table 2.** Model outputs for survival given increasing clutch size for brood manipulation (n=30 [female], 20 [male], and 8 [mixed sex]) and observational studies for the different sexes (n=11 [female], 7 [male], and 2 [mixed sex]).

Mixed-sex studies were found to be at the extremes of the trend, a reflection of species who lay smaller clutch sizes rather than an effect of the mixed sex itself. Values in bold show statistical significance of $p < 0.05$.

| Clutch size measure | | Sex | Estimate | SE | p Value | CI.lb | CI.ub |
|---|---|---|---|---|---|---|---|
| | | Female | −0.0361 | 0.0407 | 0.3754 | −0.1158 | 0.0437 |
| | | Male | 0.0186 | 0.0452 | 0.6807 | −0.07 | 0.1072 |
| | Brood manipulation | Mixed | −0.2079 | 0.0617 | **0.0007** | −0.3287 | −0.087 |
| | | Female | 0.1279 | 0.104 | 0.2189 | −0.076 | 0.3317 |
| | | Male | 0.0232 | 0.1067 | 0.8282 | −0.186 | 0.2324 |
| Raw | Observational | Mixed | 0.49 | 0.2638 | 0.0632 | −0.027 | 1.007 |

*Appendix 1—table 2 Continued on next page*

*Appendix 1—table 2 Continued*

| Clutch size measure | | Sex | Estimate | SE | p Value | CI.lb | CI.ub |
|---|---|---|---|---|---|---|---|
| | | Female | –0.082 | 0.0665 | 0.2179 | –0.2124 | 0.0484 |
| | | Male | 0.0318 | 0.0764 | 0.6778 | –0.1181 | 0.1816 |
| | Brood manipulation | Mixed | –0.2686 | 0.1264 | **0.0336** | –0.5163 | –0.0209 |
| | | Female | 0.1732 | 0.1491 | 0.2452 | –0.119 | 0.4654 |
| | | Male | 0.0076 | 0.156 | 0.9614 | –0.2982 | 0.3133 |
| Standardised | Observational | Mixed | 0.5216 | 0.3214 | 0.1047 | –0.1084 | 1.1516 |
| | | Female | –0.2776 | 0.1909 | 0.1458 | –0.6517 | 0.0965 |
| | | Male | 0.1135 | 0.2368 | 0.6318 | –0.3506 | 0.5776 |
| | Brood manipulation | Mixed | –0.6169 | 0.2773 | **0.0261** | –1.1604 | –0.0734 |
| | | Female | 0.5721 | 0.3148 | 0.0692 | –0.0449 | 1.1892 |
| | | Male | 0.0639 | 0.3287 | 0.8459 | –0.5803 | 0.7081 |
| Mean adjusted | Observational | Mixed | 0.9363 | 0.5614 | 0.0953 | –0.164 | 2.0366 |

Model = observational_or_experimental + sex, random = (species, phylogeny, study reference).

**Appendix 1—table 3.** $I^2$ values for each model showing the proportion of variation accounted for by the random effects of the model.

The phylogenetic signal was included as a correlation matrix within the model.

| Model | $I^2$ | | | | |
|---|---|---|---|---|---|
| | Total | Species | Phylogenetic | Reference | Total species effect (species + phylogenetic) |
| Raw | 0.494 | 0.000000003 | 0.287 | 0.208 | 0.287 |
| Standardised | 0.542 | 0.080 | 0.00000002 | 0.463 | 0.080 |
| Proportional | 0.428 | 0.137 | 0.00000001 | 0.291 | 0.137 |

**Appendix 1—table 4.** Excluded studies and the rationale for exclusion.

| Reference | Species | Reason for exclusion |
|---|---|---|
| *Ashcroft, 1979* | *Puffinus puffinus* | No parental survival values given clutch/brood size. Also no clutch/brood size variation in focal species. |
| *Erikstad et al., 2009* | *P. puffinus* | No clutch/brood size manipulation. Manipulation is age of offspring. |
| *Wernham and Bryant, 1998* | *P. puffinus* | No clutch/brood size variation in study. |
| *Wiebe, 2005* | *Colaptes auratus* | Mate removal, not clutch/brood manipulation. |
| *Askenmo, 1979* | *Ficedula hypoleuca* | Doesn't state manipulation size. |
| *Tinbergen and Both, 1999* | *Parus major* | Manipulation is to equalise brood size throughout population. |
| *Annett and Pierotti, 1999* | *Larus occidentalis* | Breeding lifespan not survival. |
| *Murphy, 2007* | *Tyrannus tyrannus* | No survival values given. |

*Appendix 1—table 4 Continued on next page*

*Appendix 1—table 4 Continued*

| Reference | Species | Reason for exclusion |
| --- | --- | --- |
| *Lessells, 1986* | *Branta canadensis* | No parental survival values given clutch/brood size. |
| *Schaub and von Hirschheydt, 2009* | *Hirundo rustica* | Clutch sizes are pooled (0 offspring,1–6 offspring and 6+ offspring) with large variation in each group meaning it is not informative for our study to gain reasonably accurate survival given clutch size raised. |
| *Milonoff and Paananen, 1993* | *Bucephala clangula* | Clutch size before manipulation varies significantly. |
| *Blondel et al., 1998* | *Parus caeruleus* | No parental survival values given clutch/brood size. |
| *Knowles et al., 2010* | *P. caeruleus* | No parental survival values given clutch/brood size. |
| *Kluijver, 1971* | *P. major* | Combined first and second broods. |

## Appendix 2

### Methods details of brood manipulation data extraction

Firstly, we re-extracted the raw parental survival values for each given clutch size from the studies in *Santos and Nakagawa, 2012*. This gave a continuous scale of clutch size, as opposed to *Santos and Nakagawa, 2012* who compared both brood increases and brood reductions, irrespective of the size of these manipulations, as combined categories to the control category of a study. This allowed us to directly compare observational and experimental studies and allowed us to account for the severity of the change in clutch size. We then expanded the number of studies by also including mixed-sex studies, where survival returns were combined for both parents, and included studies published in the years following publication of the *Santos and Nakagawa, 2012* paper. To do this we used a key word search on Web of Science and Google Scholar using the following terms: "longevity" OR "lifespan" OR "survival" AND "breeding success" OR "brood size" OR "clutch size" OR "number of chicks" OR "number of eggs" AND "trade-off" OR "trade offs" AND fitness AND life-history AND avian OR bird OR birds OR ornithology.

### Details of equations used to calculate selection differentials

*Equation 1*: Simple exponential survival approximation function for when risk is equal, that is, when annual survival independent of age used. Where $S$ = survival, $t$ = age, and lambda ($\lambda$) is annual survival.

Annual survival in the treatment category is changed by increased reproductive effort on a log odd ratio scale. To adjust for these survival consequences, we first convert this to a linear scale.

$$S\left(t\right) = e^{-\lambda t} \tag{1}$$

*Equation 2*: Conversion of logged OR (ES) function of $R$ (reproductive effort, as added chicks) to a linear OR for an added chick. The ES is estimated from meta-analysis and a range is tested to explore parameter space.

$$OR_R = e^{\left(ES*R\right)} \tag{2}$$

*Equation 3*: Lifetime reproductive success. Combining *Equation 1 and 2* gives a survival function adjusted for the OR adjusted for reproductive effect (right-hand side). When we multiply the adjusted survival function by reproductive effort at each $t$ and sum this, we get total LRS. Reproductive effort here is the extra reproductive output ($R$) together with the focal species brood size ($R_{base}$). Note we omit the first year when survival is 1, that is, $t$ = 0. This means we assume reproductive effort happens in $t$ = 1 and has direct consequences, that is, individuals die during reproduction and lose their brood.

$$LRS = \sum_{t=1}^{\infty} \left(R_{base} + R\right) * e^{-\lambda * OR_R * t} \tag{3}$$

### Details of selection effect estimates

We expressed fitness consequences as a proportional change in lifetime reproductive success (*Figure 4*). To compare the selection differentials expressed as the standard deviation from the mean, we first calculated the linear difference in fitness per egg. This is relevant as selection has the potential to drive rapid evolution from a phenotypic change of 0.1–0.3 SD in a generation (*Cohen et al., 2020*). We assumed a binary population with half producing an extra egg. Note, however, that this probably leads to an overestimate of the fitness effect; the proportion in the population genetically predisposed for this trait is most probably smaller than half. We can then use estimates of the SD to investigate whether rapid selection is expected. Only furthest away from the observed fast–slow pace-of-life continuum (the diagonal of *Figure 3*) was selection strong enough to come close to this rule of thumb. At an average clutch size of 2, a survival rate of 0.2 and using the meta-analytic effect size observed, –0.05, selection (assuming $h^2$ of 1, thus probably overestimating the change in one generation) would result in a 0.09 SD change in the phenotypic mean over one generation. For this example, the species' average clutch size would rapidly evolve to be larger. Over several generations, the selection effect would reduce to the point where the species would lie where an extant species is observable – a species with a survival rate of 0.2 and a species'

average clutch size of 10. At this point, the selection effect reduces to 0.007 SD change from the phenotypic mean over one generation. This selection effect is small and therefore, at a level where environmental and genetic effects could counterbalance (or even reverse periodically) any selection pressures, maintaining the constraint of clutch size to the within-species variation.

