## [Editor Report · eLife Assessment]

This **important** study challenges conventional life-history theory by demonstrating that reproductive-survival trade-offs are minimal in birds, except when reproductive effort is experimentally exaggerated. The evidence is **solid**, drawing from a meta-analysis of over 30 bird species, and effectively separates the effects of individual quality from reproductive costs. The findings will be of broad interest to evolutionary biologists and ecologists studying life-history trade-offs and reproductive strategies.

---

## [Referee Report · Reviewer #4 (Public review)]

Summary:

This is an important study that underscores that reproduction-survival trade-offs are not manifested (contrary to what generally accepted theory predicts) across a range of studies on birds. This has been studied by a meta-analytical approach, gathering data from a set of 46 papers (30 bird species). The overall conclusion is that there are no trade-offs apparent unless experimental manipulations push the natural variability to extreme values. In the wild, the general pattern for within-species variation is that birds with (naturally) larger clutches survive better.

Likely impact:

I think this is an important contribution to a slow shift in how we perceive the importance of trade-offs in ecology and evolution in general. While the current view still is that one individual excelling in one measure of its life history (i.e. receiving benefits) must struggle (i.e. pay costs) in another part. However, a positive correlation between all aspects of life history traits is possible within an individual (such as due to developmental conditions or fitting to a particular environment). Simply, some individuals can perform generally better (be of good quality than others).

---

## [Author Response]

The following is the authors’ response to the previous reviews.

**Public Reviews:**

**Reviewer #4 (Public review):**

We would like to thank the reviewer for their careful consideration of our manuscript. The suggestions have been useful in improving our manuscript. Please see our responses to the specific comments below.

Summary:This is an important study that underscores that reproduction-survival trade-offs are not manifested (contrary to what generally accepted theory predicts) across a range of studies on birds. This has been studied by a meta-analytical approach, gathering data from a set of 46 papers (30 bird species). The overall conclusion is that there are no trade-offs apparent unless experimental manipulations push the natural variability to extreme values. In the wild, the general pattern for within-species variation is that birds with (naturally) larger clutches survive better.Strengths:I agree this study highlights important issues and provides good evidence of what it claims, using appropriate methods.Weaknesses:I also think, however, that it would benefit from broadening its horizon beyond bird studies. The conclusions can be reinforced through insights from other taxa. General reasoning is that there is positive pleiotropy i.e. individuals vary in quality and therefore some are more fit (perform better) than others. Of course, this is within their current environment (biotic, abiotic, social. ...), with consequences of maintaining genetic variation across generations - outlined in Maklakov et al. 2015 (https://doi.org/10.1002/bies.201500025). This explains the outcomes of this study very well and would come to less controversy and surprise for a more general audience.I have two fish examples in my mind where this trade-off is also discounted. Of course, given that it is beyond brood-caring birds, the wording in those studies is slightly different, but the evolutionary insight is the same. First, within species but across populations, Reznick et al. (2004, DOI: 10.1038/nature02936) demonstrated a positive correlation between reproduction and parental survival in guppies. Second, an annual killifish study (2021, DOI: 10.1111/1365-2656.13382) showed, within a population, a positive association between reproduction and (reproductive) aging.In fruit flies, there is also a strong experimental study demonstrating the absence of reproduction-lifespan trade-offs (DOI: 10.1016/j.cub.2013.09.049).I suggest that incorporating insights from those studies would broaden the scope and reach of the current manuscript.

We would like to thank the reviewer for this useful insight and for highlighting these studies. We have added detail in our discussion around positive correlations observed in the wild, and how positive pleiotropy has been presented as an explanation. We have also added the suggested studies as references to demonstrate the reproduction-lifespan trade-off has been shown to be absent. See lines 257-260.

Likely impact:I think this is an important contribution to a slow shift in how we perceive the importance of trade-offs in ecology and evolution in general. While the current view still is that one individual excelling in one measure of its life history (i.e. receiving benefits) must struggle (i.e. pay costs) in another part. However, a positive correlation between all aspects of life history traits is possible within an individual (such as due to developmental conditions or fitting to a particular environment). Simply, some individuals can perform generally better (be of good quality than others).

We would like to thank the reviewer for highlighting the importance of our study. We hope our study will help the research community reflect on the importance of trade-offs between life-history traits and consider other possible explanations as to why variation in life-history traits is maintained within species.